# External validation of VO$_{2max}$ prediction models based on recreational and elite endurance athletes

**Szczepan Wiecha**[1]*, **Przemysław Seweryn Kasiak**[2], **Igor Cieśliński**[1], **Tim Takken**[3], **Tomasz Palka**[4], **Beat Knechtle**[5], **Pantelis T. Nikolaidis**[6], **Łukasz A. Małek**[7], **Marek Postuła**[8], **Artur Mamcarz**[9], **Daniel Śliż**[9]

**1** Faculty in Biala Podlaska, Department of Physical Education and Health, Jozef Pilsudski University of Physical Education in Warsaw, Biala Podlaska, Poland, **2** 3rd Department of Internal Medicine and Cardiology, Students' Scientific Group of Lifestyle Medicine, Medical University of Warsaw, Warsaw, Poland, **3** Department of Medical Physiology, Child Development & Exercise Center, Wilhelmina Children's Hospital, UMC Utrecht, Utrecht, The Netherlands, **4** Faculty of Physical Education and Sport, Department of Physiology and Biochemistry, University of Physical Education in Krakow, Krakow, Poland, **5** Institute of Primary Care, University of Zurich, Zurich, Switzerland, **6** School of Health and Caring Sciences, University of West Attica, Athens, Greece, **7** Department of Epidemiology, Cardiovascular Disease Prevention and Health Promotion, National Institute of Cardiology, Warsaw, Poland, **8** Department of Experimental and Clinical Pharmacology, Centre for Preclinical Research and Technology, Medical University of Warsaw, Warsaw, Poland, **9** 3rd Department of Internal Medicine and Cardiology, Medical University of Warsaw, Warsaw, Poland

* szczepan.wiecha@awf.edu.pl

**Data Availability Statement:** All relevant data are within the paper and its Supporting Information files.

## Abstract

In recent years, numerous prognostic models have been developed to predict VO2max. Nevertheless, their accuracy in endurance athletes (EA) stays mostly unvalidated. This study aimed to compare predicted VO2max (pVO2max) with directly measured VO2max by assessing the transferability of the currently available prediction models based on their R$^2$, calibration-in-the-large, and calibration slope. 5,260 healthy adult EA underwent a maximal exertion cardiopulmonary exercise test (CPET) (84.76% male; age 34.6±9.5 yrs.; VO2max 52.97±7.39 mL·min$^{-1}$·kg$^{-1}$, BMI 23.59±2.73 kg·m$^{-2}$). 13 models have been selected to establish pVO2max. Participants were classified into four endurance subgroups (high-, recreational-, low- trained, and "transition") and four age subgroups (18–30, 31–45, 46–60, and ≥61 yrs.). Validation was performed according to TRIPOD guidelines. pVO2max was low-to-moderately associated with direct CPET measurements ($p$>0.05). Models with the highest accuracy were for males on a cycle ergometer (CE) (Kokkinos R$^2$ = 0.64), females on CE (Kokkinos R$^2$ = 0.65), males on a treadmill (TE) (Wasserman R$^2$ = 0.26), females on TE (Wasserman R$^2$ = 0.30). However, selected models underestimated pVO2max for younger and higher trained EA and overestimated for older and lower trained EA. All equations demonstrated merely moderate accuracy and should only be used as a supplemental method for physicians to estimate CRF in EA. It is necessary to derive new models on EA populations to include routinely in clinical practice and sports diagnostic.

**Funding:** The authors received no specific funding for this work.

**Competing interests:** The authors have declared that no competing interests exist.

## Introduction

The concept of maximal oxygen uptake ($VO_{2max}$) was first suggested by Hill et al. in the 1920s [1]. $VO_{2max}$ is the highest attained oxygen uptake during an incremental exercise test with large muscle groups (e.g., treadmill or cycling). $VO_{2max}$ is an important parameter to objectively assess cardiorespiratory fitness (CRF) both in healthy people and those suffering from cardiovascular diseases (CVD) [2,3]. The American Heart Association (AHA) recognized that CRF, described mainly as a $VO_{2max}$, should be used as an essential factor in the comprehensive diagnostic process [3]. Moreover, a lower level of CRF is strongly related to a higher risk of CVDs, death from numerous cancer types, and all-cause mortality [4]. This represents a switch from risk factors widely discussed in recent decades, such as smoking, hypertension, or hyperlipidemia [2,3,5].

### $VO_{2max}$ in sports & performance diagnostics

$VO_{2max}$ is an important variable in endurance sports, such as running, cycling, swimming, triathlon, or team sports [6]. $VO_{2max}$ strongly correlates with athlete's aerobic performance, could be applied to prescribe training properly, and is useful to assess adaptation to exercise [7–9]. Furthermore, the $VO_{2max}$ could help in the prediction of a race time [10,11]. Elite athletes achieve varied $VO_{2max}$ values, dependent on their discipline and training experience [12,13]. Males typically have higher $VO_{2max}$ than females [14], and $VO_{2max}$ values decrease with age [15]. Body weight and height are related as well as the testing mode. Higher $VO_{2max}$ values are observed on a treadmill compared to the cycle ergometer [16]. Briefly, Kaminky et al. indicate the level of training, the method of testing (cycle ergometry, treadmill, rowing machine, etc.), co-existing CVDs, and respiratory exchange ratio (RER) as factors influencing $VO_2$ [17]. Other contributors perhaps include the psychological attitude of the athlete to the effort (i.e. that the CPET conducting until refuse may potentially last longer), race, and ethnicity [18,19]. Due to its numerous practical implications and variability, it is important to precisely assess $VO_{2max}$ in different athletic disciplines and populations [20].

### $VO_{2max}$ in clinical practice

Measuring $VO_{2max}$ is also especially important under clinical conditions during the examination of the cardiovascular system [3,21]. It could be regarded as the integrated function of (amongst others) lungs, heart, blood vessels, and muscles [9,22]. Recommendations for $VO_{2max}$-testing include the presence of ambiguous pathologic exertional symptoms, cardiovascular risk estimation, and monitoring response to applied treatment [21]. Moreover, understanding the exercise limitation is crucial information for healthcare professionals to monitor cardiac status and could be used to prescribe treatment properly for those suffering from CVDs [2,23]. Therefore, $VO_{2max}$ is a practically relevant parameter for a new, growing population of patients in cardiologic ambulatory care- endurance athletes (EA) [21]. Both highly trained endurance athletes (HTEA), recreational endurance athletes (REA) and low-trained endurance athletes (LTEA) with suspected CVD and those undergoing cardiopulmonary exercise testing (CPET) for periodic training evaluation are potential candidates for $VO_{2max}$ assessment [21].

### Epidemiology of CVDs among EA

In light of current literature, there is growing importance in preventing and treating CVDs in athletes. As the number of people practicing endurance sports increases, new patient populations arose, professional and former EAs [3]. For example, in recent times, due to the SARS-CoV-2 infection, some athletes have had cardiac involvement. CPET and $VO_{2max}$ assessment

are important elements of a comprehensive diagnostic approach [24]. To deepen the epidemiological data, it is worth mentioning that arterial hypertension is the most common CVD among physically active people. The risk of CVD is found especially in the group of people after 35 years old (thus former and retired EA). Although sport is recognized as a preventive factor for CVD, Medical Practitioners should be aware of the prevalence of risk factors among EA [25]. What is more, as claimed by Petek et al. among the wide cohort of collegiate athletes prevalence of persistent or exertional symptoms on return to exercise occurs only in 44/3529 (1.2%). This has been achieved, among others, by a properly conducted diagnostic and screening process consisting of CPET and $VO_2$ assessment [21].

Again, as observed by Petek et al. the comparison of $VO_{2max}$ with cardiac morphology and echocardiography may facilitate the correct planning of the therapy [24]. Moreover, Moulson et al. recently found that CPET is a valuable component of the Return to Play Program and cardiac screening in young competitive EAs following SARS-CoV-2 infection [26]. To summarize, directly measured $VO_{2max}$ can be used as a valuable predictive cardiometabolic risk factor.

### CPET protocol and applicability of prediction formulae

The gold standard to measure $VO_{2max}$ is performing a CPET [27]. $VO_{2max}$ is reached when the subject meets the physiological limit and maintains it for some time (usually 15-s, 30-s, or 60-s) [28]. Due to practical reasons, such as high costs of the procedure or a lack of testing devices as well as health contraindications, this form of measuring is often not possible to apply in a sports setting [27].

Parameters such as age, sex, and heart rate (HR) could be used to predict $VO_{2max}$ through various models [27,29]. The reliability of this potentially non-sophisticated and valuable method is complicated and doubtful because of low accuracy, especially in women, extremely small or tall subjects, and in individuals with high BMI values [30,31]. In the 2013 statement, AHA pointed out that there is a need for a universal and transferable prediction standard [32].

Prediction formulae undoubtedly have numerous advantages, however, those currently used were created on different populations and with the incorporation of heterogeneous testing modes [33]. Indeed, proper external validation should be a mandatory stage before the new model will be widely used [34,35]. Moreover, the risk of using only predicted values is a certain inaccuracy and error in the particular equations [36]. On the other hand, the benefit is that there is no need to undergo full CPET, which may be expensive, or when there is limited availability of specialized clinics, equipment, etc (eg. in a field settings) [37,38].

Validation studies are performed to evaluate a given model in varied conditions and on differentiated populations to assess its possible measurement bias and the ability to extrapolate its results [39]. This study aimed to externally evaluate prediction formulae on EA tested under the same conditions from one tertiary care sports diagnostic center. EAs were selected for a study population as $VO_{2max}$ is an important parameter in the evaluation of the overall fitness level and the selected equations are often derived from the athletic population [22]. The secondary aim was to assess the impact of age and CRF on the risk of error and bias in tested models. We hypothesize that their validity may not be sufficient to make them an equivalent method for directly measured $VO_{2max}$.

## Material and methods

We applied TRIPOD guidelines for the development and validation of prediction models (for detailed protocol see Supplementary information. TRIPOD Checklist for Prediction Model Validation) [39]. Results from CPETs collected between 2013–2021 were retrospectively analyzed. Maximal-effort examinations consisted of the treadmill (TE) or the cycle ergometry

(CE) tests, paired with body composition (BC) analysis took place in the medical clinic (www.sportslab.pl, Warsaw, Poland). Tests were performed on an individual request as a part of regular endurance assessment or training monitoring.

## Cardiopulmonary exercise testing protocol

Cardiopulmonary exercise tests (CPET) were preceded by body mass (BM) and fat mass (FM) analysis with 5 kHz/50 kHz/250 kHz electrical bioimpedance method on the body composition (BC) monitor (Tanita, MC 718, Japan. Conditions during BC and CPET were: 40 $m^2$ indoor, air-conditioned area, 40–60% humidity, temperature 20–22˚C, altitude 100 m MSL. Endurance athletes (EA) were instructed via e-mail on how to prepare: avoid any demanding exercises 24 hours before CPET, consume a high carbohydrate meal and hydrate with isotonic beverages 2–3 hours earlier, and exclude any stimulants or caffeine on the day of the procedure.

Cycle ergometry (CE) examination was performed on a cycle ergometry Cyclus-2 (RBM elektronik-automation GmbH, Leipzig, Germany) and treadmill (TE) examination was conducted on a mechanical treadmill (h/p/Cosmos quasar, Germany). CPET scores were measured using a Hans Rudolph V2 Mask (Hans Rudolph, Inc, Shawnee, KS, USA), a gas exchange analyzer Cosmed Quark CPET (Rome, Italy), and dedicated manufacturer's software (from PFT Suite to Omnia 10.0E.). Data collection was performed with a breath-by-breath acquisition system and a 15-s filter was used for data analysis. Each breath was considered as a separate point and all points were included in the calculation of the average $VO_{2max}$ value.HR was measured via ANT and a torso strap as a part of the Cosmed Quark set (product accuracy comparable to ECG; ± 1 bpm.). The CPET device was calibrated with reference gas (16% $O_2$; 5% $CO_2$) and turbine flow for each person separately, according to manufacturer recommendations. Equipment software was regularly actualized between 2013–2021. Three gas analyzing devices were utilized and each one has been changed after 36–48 months. Every part of CPET equipment was periodically verified by manufacturer employees to keep their mechanical certificates valid. Blood lactate (LA) was assessed with the usage of Super GL2 analyzer (Müller Gerätebau GmbH, Freital, Germany). The instrument was also individually prepared before each round of analysis and calibrated with reference solution before each sample set.

Exercises begin with a 5-min. warm-up (walking or pedaling with minimal resistance). Participants' endurance capacities were used to assess starting load. The initial power for CE was 60-150W and was increased in 2 min. intervals by 20-30W. The initial speed for TE was 7–12 km·h$^{-1}$ (described by a person as a "conversation pace") at 1% inclination. The pace was raised by 1 km·h$^{-1}$ every 2 min. Observer verbally encouraged athletes to keep effort as long as possible due to assess their endurance most exactly. Achievement of oxygen uptake ($VO_2$) or heart rate (HR) plateau, or volitional inability to maintain intensity were reasons for test termination. LA was measured by taking a 20 µL blood sample from a fingertip: directly prior to exercises, after any resistance or pace modification, and 3 min. after termination. Samples were obtained without an interruption in CE and TE tests. Before a proper sample was obtained, the first drops were gathered in a swab. HR (not averaged) was recorded at the highest point during intervals and used in further analysis [40]. Maximal oxygen uptake ($VO_{2max}$) was defined as an averaged maximum oxygen uptake during the 15-s period at the end of the CPET.

## Derivation cohort

The rigorous inclusion/exclusion process was applied to narrow the validation group to only those EAs who achieved maximum exertion during CPET and were free of any possible $VO_{2max}$ alleviating factors (see Fig 1. Flowchart of the inclusion-exclusion and further groups classification process).

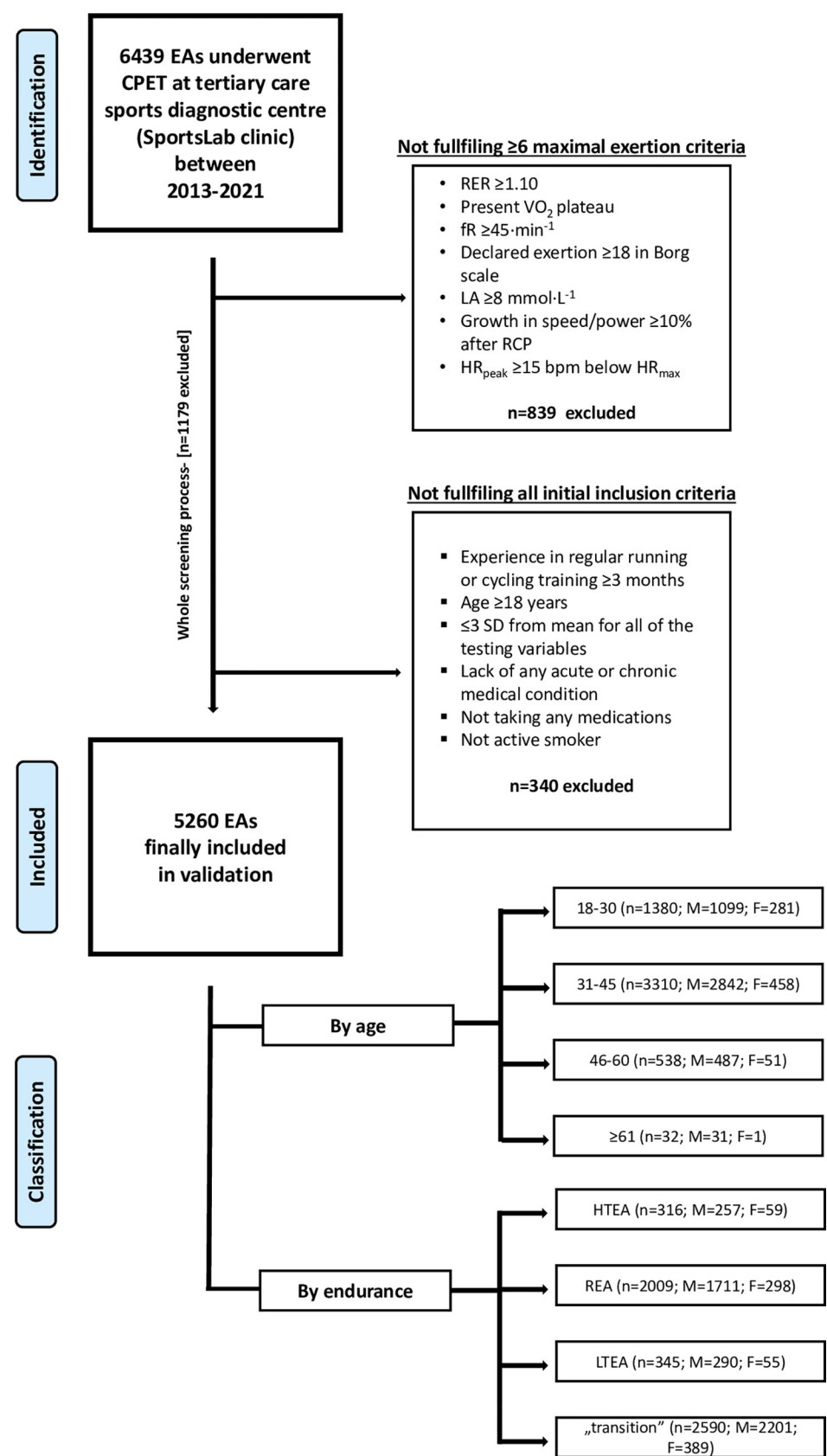

**Fig 1. Flowchart of the inclusion-exclusion and further groups classification process.** Age classification is presented in years. Endurance classification has been performed based on speed/power at respiratory compensation point (RCP) which is currently described as a variable most closely corresponding to the critical power. Moreover, the selection of a variable different from $VO_{2max}$ to the classification of participants in terms of their endurance capacity, enable to make group assignments independent of the factor directly validated in the study. Abbreviations: EA, endurance athlete; CPET, cardiopulmonary exercise testing; SD, standard deviation; RER, respiratory exchange ratio; $VO_2$, oxygen uptake (mL·min$^{-1}$·kg$^{-1}$); LA, lactate concentration (mmol·L$^{-1}$); fR, breathing frequency (breaths·min$^{-1}$); RCP, respiratory compensation point; $HR_{peak}$, peak heart rate during CPET (bpm); $HR_{max}$, maximal heart rate during CPET (bpm); F, female; M, male; HTEA, high-trained endurance athletes; REA, recreational endurance athletes, LTEA, low-trained endurance athletes.

6,439 EAs underwent CPET. Participants were eligible for preliminary inclusion if they had: (1) experience in regular running or cycling training ≥3 months, (2) age ≥18 years, (3) ≤±3 standard deviations (SD) from mean for all of the testing variables (extreme outliers were excluded), (4) lack of any acute or chronic medical condition (also musculoskeletal injuries, or addictions), (5) not taking any medications, (6) not being an active smoker.

Maximum exertion in CPET was defined as fulfilment ≥6 criteria: (1) RER ≥1.10, (2) present $VO_2$ plateau (growth <100 mL·min$^{-1}$ in $VO_2$ with more increased running or cycling intensity), (3) respiratory frequency (fR) ≥45 breaths·min$^{-1}$, (4) declared exertion during CPET ≥18 in the Borg scale [31], (5) lactate concentration (LA) ≥8 mmol·L$^{-1}$, (6) growth in speed/power ≥10% of RCP after exceeding the respiratory compensation point (RCP), (7) peak heart rate ($HR_{peak}$) ≥15 bpm below predicted maximal heart rate ($HR_{max}$) [40].

Finally, 5,260 EA met all inclusion criteria. The population was divided between males and females into four age groups: 18–30; 31–45; 46–60, ≥61 years, and 4 endurance groups: HTEA, REA, LTEA, and "transition". Endurance classification was conducted based on the speed (km·h$^{-1}$) or power (W·kg$^{-1}$) at the RCP calculated independently for each sex. Speed/power at RCP was a variable-of-choice because it is currently described as a parameter most closely corresponding to the critical endurance capacity [41,42]. Moreover, the selection of a variable different from $VO_{2max}$ to the classification of participants in terms of their endurance capacity, enabled to make group assignments independent of the factor directly validated in the study. Participants with >+1.5 SD were classified as HTEA (n = 309), <+0.5SD/>−0.5SD as REA (n = 2,033), <−1.5 SD as LTEA (n = 339). To precisely distinguish endurance subgroups, those placed between ≥+0.5SD/≤+1.5SD and between ≤−0.5SD/≥−1.5SD were classified as "transition" (n = 2.579). Models' validation was conducted on each of the age and endurance cohorts independently (except the „transition" group) both for $TE_{VO2max}$ and $CE_{VO2max}$.

## Selected prediction models

Candidate models were found from previous systematic reviews for CPET testing (up to February 2019) [43,44] and additional literature search in PubMed, MEDLINE, EMBASE, Scopus, and Web of Science databases (for a period between March 2019- December 2021 and meta-analyses) for keywords: Cardiopulmonary exercise testing, Cardiorespiratory fitness, Exercise testing, $VO_{2max}$, $VO_{2peak}$.

Exclusion criteria were: (1) not reporting $VO_{2max}$ parameters, (2) usage of other ergometers than CE or TE during CPET, (3) consideration of parameters not possible to verify in our sample (declared physical activity level, time to exhaustion), (4) generating unviable results multiple times (<0 or >100 mL·min$^{-1}$·kg$^{-1}$ $VO_{2peak}$), (5) being derived from pediatric (the oldest participant <18 years old) or geriatric population (the youngest participant ≥61 yrs.), (6) being derived before 01.01.2000, (7) not reporting $R^2$ from internal/external validation, (8) being derived from group <1000 participants (for the general population) or <200

participants (specifically for EA population), (9) methodological quality <7 points according to ATS/ACCP guidelines [37], (10) usage of other testing technique than breath by breath, (11) CPET protocol not carried out following the recommended clinical ATS/ACCP guidelines [45].

Moreover, the Wasserman et al. [22] model was validated in the study due to its well-established reputation. Equations from 2 meta-analyses [46,47] were also considered because of their wide range of applications for EA.

During studies selection, we did not define the criteria for the VO2max measurement protocol due to the high variability of the currently described methods. Nevertheless, according to the current literature, different testing protocols were applied for runners and cyclists [48–51]. Similar values of $VO_{2max}$ were observed, which suggests that it is possible to provide an exact comparison between them.

13 equations from 8 different publications were included in the analysis. Their detailed characteristics are presented in the supplementary material (S2 File).

## Statistical analysis

Baseline statistics were exported into the Excel file (Microsoft Corporation, Washington, USA) and are presented as mean (±SD and 95% CI) or frequency (percentage) for categorical variables, and median for continuous variables. Differences between subgroups (all continuous variables) were analyzed using the ANOVA test-of-variance and post-hoc HSD Tukey test. There was not any missing data in the whole population. Thus, an entire cohort has been validated.

External validation was conducted by following the recommendations for the validation and interpretation of diagnostic prediction models [34]. In summary, we assessed equations accuracy by comparisons between the originally established formulas and data obtained directly from CPETs and BC examinations (e.g., $VO_{2max}$, BMI). Linear model regressing measured $VO_2max$ on $pVO_2max$ was generated for each equation. Performance considered as the proximity of the observed and expected CRF, was evaluated with the usage of the $R^2$, root mean square error (RMSE). Cutoffs for $R^2$ were: (1) $R^2 < 0.3$ for none or very weak effect size; (2) $0.3 < R^2 < 0.5$ for weak or low effect size; (3) $0.5 < R^2 < 0.7$ for moderate effect size; (4) $R^2 > 0.7$ for high effect size [52]. Additionally, calibration slope (the slope of a linear regression model that includes the model's linear predictor as the only covariate parameter estimate where 1 being ideal; C1), and calibration-in-the-large (mean observed compared to mean predicted value where 0 being ideal; C2) were calculated.

Ggplot 2 package in RStudio (R Core Team, Vienna, Austria; version 3.6.4), originally written Python script (Python Software Foundation, Delaware, USA; version 3.10.1), and STATA software (StataCorp, College Station, Texas, USA; version 15.1) were used in statistical analysis. The significance borderline was at a two-sided $p$-value <0.05.

## Ethical approval

All parts of the study were approved by the Bioethical Committee-IRB of the Medical University of Warsaw (AKBE/32/2021) and were conducted in line with the Declaration of Helsinki. Moreover, each EA has to provide their written consent in a separate document.

## Results

From a total of 6,439 endurance athletes (EA) who underwent CPET at a tertiary care sports diagnostic center in Poland, 5,260 EA met the inclusion criteria. Participants' basic anthropometric characteristics are shown in Table 1. The average age of was 35.04±9.58 yrs. in the male

**Table 1. Participants' basic anthropometric characteristics.**

| Variable | Male [n = 4459; 84.76%] | Female [n = 801; 15.24%] |
|---|---|---|
| **Baseline characteristic** | | |
| **Age (years)** | 35.04 (9.58)* | 32.25 (8.99) |
| **Height (cm)** | 179.42 (6.60)* | 167.19 (6.88) |
| **Weight (kg)** | 77.23 (10.32)* | 60.60 (8.73) |
| **BMI (kg·m$^{-2}$)** | 23.95 (2.63)* | 21.64 (2.38) |
| **BF (%)** | 15.68 (4.55)* | 22.04 (5.46) |
| **FFM (kg)** | 64.87 (7.17)* | 47.08 (6.36) |
| **Endurance groups characteristic** | | |
| **HTEA (n = 316; 6.08%)** | 257 (4.89) | 59 (1.12) |
| **REA (n = 2009; 38.19%)** | 1711 (32.52) | 298 (5.67) |
| **LTEA (n = 345; 6.56%)** | 290 (5.51) | 55 (1.05) |
| **„transition" (n = 2590; 49.24%)** | 2201 (41.84) | 389 (7.40) |
| **Age groups characteristic** | | |
| **Age 18–30 (n = 1380; 26.24%)** | 1099 (20.89) | 281 (20.89) |
| **Age 31–45 (n = 3310; 62.92%)** | 2842 (54.03) | 458 (8.71) |
| **Age 46–60 (n = 538; 10.23%)** | 487 (9.26) | 51 (0.97) |
| **Age >61 (n = 32; 0.61%)** | 31 (0.59) | 1 (0.02) |

Abbreviations: CE, cycle ergometry; TE, treadmill; BMI, body mass index; BF, body fat; FFM, fat-free mass; HTEA, high-trained endurance athletes; REA, recreational endurance athletes; LTEA, low-trained endurance athletes. Continuous value is presented as mean (SD), while categorical was showed as numbers (%) when appropriate. Comparisons between subgroups (*p* value) were obtained by one-way ANOVA, Student's t-test, and post-hoc HSD Tukey test. Significant differences (*p<0.05*) were marked with [*].

population (n = 4,459; 84.76%) and 32.25±8.99 yrs. in the female population (n = 801; 15.24%).

CPET data are shown in the supplementary material (S3 File). For male EA, observed VO$_{2max}$ was significantly higher for TE (n = 3,330) than for CE (n = 1,129) (54.10±6.93 vs 51.92±8.05 mL·min$^{-1}$·kg$^{-1}$; p<0.05). In female athletes, VO$_{2max}$ was similar for TE (n = 671) and for CE (n = 130) (48.79±6.67 vs 49.05±6.64 mL·min$^{-1}$·kg$^{-1}$; (p>0.05). HTEA had significantly higher (p<0.05) levels of VO$_{2max}$, the speed at RCP (S$_{RCP}$), and the power at RCP (P$_{RCP}$). Observed VO$_{2max}$ was significantly lower (p<0.05) in the LTEA subgroup.

Briefly, VO$_{2max}$ differed significantly between the selected equations. The performance of prediction models is presented in Tables 2 and 3 along with R$^2$, root mean square error (RMSE), calibration-in-the-large (C1), and calibration slope (C2). Figs 2–5 shows the regression analysis of observed vs predicted VO$_{2max}$ stratified by age for the whole population, HTEA, REA, and LTEA, respectively. Subgroups that did not meet the TRIPOD guidelines [30] to consider their validation results as reliable (i.e., n≥100) were additionally marked in tables and graphs.

Performance calculations for the whole population and each subgroup, with comparison (mean and SD) between observed and predicted VO$_{2max}$ are presented in the supplementary material (Table 3a–3d in S4 File). For TE, the lowest non-significant differences (mean and CI) were for Petek's equation both in males (mean = –0.11; CI, –0.42, 0.20) and females (mean = –0.52; CI,–1.20, 0.16). For CE, the lowest non-significant differences (mean and CI) were for Petek's equation in the male population (mean = –0.08; CI, –0.68, 0.52). Similarly, for the female population, the lowest but significant differences were also for Petek's equation

**Table 2. Performance of selected models stratified by endurance level and sex.**

**Prior Equations in CE**

| Validated subgroup | Males HTEA‡ [n=69] R² | RMSE | C1 | C2 | Males REA [n=429] R² | RMSE | C1 | C2 | Males LTEA‡ [n=81] R² | RMSE | C1 | C2 | Females HTEA‡ [n=10] R² | RMSE | C1 | C2 | Females REA‡ [n=57] R² | RMSE | C1 | C2 | Females LTEA‡ [n=11] R² | RMSE | C1 | C2 |
|---|---|---|---|---|---|---|---|---|---|---|---|---|---|---|---|---|---|---|---|---|---|---|---|---|
| Wilson et al.† (mL·min⁻¹·kg⁻¹) | 0.22* | 6.62 | 16.59 | 0.76* | 0.24* | 4.74 | 11.20* | 0.68* | 0.21* | 4.37 | 8.03 | 0.53* | n/a | n/a | n/a | n/a | n/a | n/a | n/a | n/a | n/a | n/a | n/a | n/a |
| Fitzgerald et al.† (mL·min⁻¹·kg⁻¹) | n/a | n/a | n/a | n/a | n/a | n/a | n/a | n/a | n/a | n/a | n/a | n/a | -0.01 | 3.38 | 40.78* | 0.30 | 0.46* | 4.42 | 44.93* | 0.08 | 0.46* | 4.88 | -17.15 | 1.05* |
| Wasserman et al. (mL·min⁻¹) | 0.27* | 446 | 2064.6* | 0.88* | 0.48* | 363 | 1390.49* | 0.93* | 0.60* | 351 | 1164.51* | 0.73* | 0.40* | 291 | 298.68 | 1.71* | 0.66* | 557 | 2157.49* | 0.45 | 0.66* | 314 | -1449.16 | 2.16* |
| Kokkinos et al. (1 for males/ 2 for females)§ (mL·min⁻¹·kg⁻¹) | 0.35* | 6.06 | 10.69 | 0.86* | 0.31* | 4.53 | 10.17* | 0.90* | 0.35* | 4.91 | 36.54* | 0.11 | 0.09 | 3.27 | 43.75* | 0.23 | 0.63* | 3.73 | 15.97* | 0.73* | 0.63* | 4.04 | -6.40* | 1.33 |
| Kokkinos et al. (3)§ (mL·min⁻¹·kg⁻¹) | 0.35* | 6.06 | 10.53 | 0.91* | 0.31* | 4.53 | 9.99* | 0.94* | 0.00 | 4.91 | 36.52* | 0.12 | 0.09 | 3.27 | 43.70* | 0.25 | 0.63* | 3.73 | 15.80* | 0.78* | 0.63* | 4.04 | -6.71 | 1.42* |
| Mylius et al. (mL·min⁻¹) | 0.16 | 480.53 | 2147.43* | 0.75* | 0.35 | 405.87 | 925.17* | 1.03* | 0.22 | 489.44 | 498.10 | 0.99* | 0.09 | 251.25 | 835.22 | 1.27* | 0.63* | 550.75 | 2043.54* | 0.46 | 0.63* | 412.29 | -451.49 | 1.47* |
| Petek et al. (L·min⁻¹) | 0.15* | 0.48 | 1.91* | 0.61* | 0.27* | 0.43 | 1.10* | 0.73* | 0.21* | 0.49 | 0.53 | 0.75* | 0.27 | 0.32 | 1.04 | 0.80 | 0.03 | 0.56 | 2.14* | 0.29 | 0.03 | 0.53 | 0.86 | 0.81 |

**Prior Equations in TE**

| Validated subgroup | Males HTEA [n=188] R² | RMSE | C1 | C2 | Males REA [n=1282] R² | RMSE | C1 | C2 | Males LTEA [n=209] R² | RMSE | C1 | C2 | Females HTEA‡ [n=49] R² | RMSE | C1 | C2 | Females REA [n=241] R² | RMSE | C1 | C2 | Females LTEA‡ [n=44] R² | RMSE | C1 | C2 |
|---|---|---|---|---|---|---|---|---|---|---|---|---|---|---|---|---|---|---|---|---|---|---|---|---|
| Wilson et al.† (mL·min⁻¹·kg⁻¹) | 0.15 | 5.56 | 23.59* | 0.66* | 0.14* | 5.01 | 21.75* | 0.53* | 0.09* | 5.27 | 20.23* | 0.42* | n/a | n/a | n/a | n/a | n/a | n/a | n/a | n/a | n/a | n/a | n/a | n/a |
| Fitzgerald et al.† (mL·min⁻¹·kg⁻¹) | n/a | n/a | n/a | n/a | n/a | n/a | n/a | n/a | n/a | n/a | n/a | n/a | 0.10 | 5.52 | 36.50* | 0.40* | 0.08* | 4.56 | 34.57* | 0.27* | 0.10* | 4.36 | 23.40* | 0.32* |
| Wasserman et al. (mL·min⁻¹) | 0.44* | 404 | 1461.98* | 1.14* | 0.45* | 379 | 1583.39* | 0.89* | 0.56* | 411 | 1382.82* | 0.78* | 0.64* | 450 | -1097.87* | 2.58* | 0.51* | 333 | 76.45 | 1.65 | 0.42* | 363 | -396.90 | 1.66* |
| Myers et al. (mL·min⁻¹·kg⁻¹) | 0.20* | 5.40 | 31.67* | 0.70* | 0.23* | 4.75 | 26.41* | 0.62* | 0.21* | 4.93 | 24.32* | 0.51* | 0.08* | 5.58 | 44.28* | 0.38* | 0.10* | 4.49 | 34.27* | 0.42* | 0.07* | 4.44 | 30.33* | 0.29* |
| Nevill et al. (1)§§ (mL·min⁻¹·kg⁻¹) | 0.18* | 5.47 | 42.55* | 0.45* | 0.20* | 4.84 | 35.06* | 0.42* | 0.18* | 5.03 | 31.00* | 0.35* | 0.10* | 5.52 | 43.64* | 0.38* | 0.09* | 4.52 | 36.99* | 0.34* | 0.05 | 4.48 | 33.45* | 0.20 |
| Nevill et al. (2)§§ (mL·min⁻¹·kg⁻¹) | 0.18 | 5.47 | 32.33* | 0.68* | 0.21* | 4.81 | 26.77* | 0.61* | 0.21* | 4.93 | 24.84* | 0.50* | 0.08* | 5.54 | 40.02* | 0.48* | 0.09* | 4.53 | 34.65* | 0.40* | 0.09* | 4.41 | 30.34* | 0.28* |
| Petek et al. (L·min⁻¹) | 0.18* | 0.49* | 0.32 | 1.02* | 0.23* | 0.45* | 0.26 | 0.94 | 0.23* | 0.23* | 0.94* | -0.07 | 0.53* | 0.51 | -1.32* | 1.57* | 0.30* | 0.40 | 0.28 | 0.92* | 0.01 | 0.48 | 1.78* | 0.29 |

Abbreviations: CE, cycle ergometry; HTEA, high-trained endurance athletes; REA, recreational endurance athletes; LTEA, low-trained endurance athletes; $R^2$, adjusted $R^2$; RMSE, root mean square error; C1, calibration-in-the-large; C2, calibration slope; n/a, not applicable; TE, treadmill. Comparisons between subgroups ($p$ value) were obtained by one-way ANOVA, Student's t-test, and post-hoc HSD Tukey test. Significant values ($p<0.05$) were marked as [*]. All values are presented in originally derived units for each model (original unit was added in bracket). †Fitzgerald et al. and Wilson et al. are meta-analyses exclusively for one sex. §Kokkinos et al. presents 3 equations for cycle ergometry: (1) only for males, (2) only for females, (3) for both males and females. §§Nevill et al. presents 2 equations for treadmill: (1) allometric model, (2) additive model. Subgroups that did not meet the TRIPOD guidelines to consider their validation results as reliable (i.e n≥100) were marked with [‡].

**Table 3. Performance of selected models stratified by age and sex.**

**Males — Prior Equations in CE**

| Validated subgroup | Age 18–30 [n = 228] | | | | Age 31–45 [n = 733] | | | | Age 46–60 [n = 154] | | | | Age ≥61‡ [n = 14] | | | |
|---|---|---|---|---|---|---|---|---|---|---|---|---|---|---|---|---|
| | R² | RMSE | C1 | C2 | R² | RMSE | C1 | C2 | R² | RMSE | C1 | C2 | R² | RMSE | C1 | C2 |
| Wilson et al.† (mL·min⁻¹·kg⁻¹) | 0.04* | 8.49 | −5.11 | 0.94* | 0.03* | 7.18 | 11.87 | 0.66* | −0.01 | 6.96 | 44.47* | 0.07 | 0.07 | 6.71 | −41.71 | 1.79 |
| Fitzgerald et al.† (mL·min⁻¹·kg⁻¹) | n/a | n/a | n/a | n/a | n/a | n/a | n/a | n/a | n/a | n/a | n/a | n/a | n/a | n/a | n/a | n/a |
| Wasserman et al. (mL·min⁻¹) | 0.11* | 591 | 2276.75* | 0.62* | 0.13* | 482 | 2648.22* | 0.49* | 0.10* | 476 | 2566.64* | 0.49* | 0.25* | 537 | −24.81 | 1.63* |
| Kokkinos et al. (1 for males/ 2 for females)§ (mL·min⁻¹·kg⁻¹) | 0.67* | 4.94 | 14.78* | 0.83* | 0.59* | 4.66 | 15.81* | 0.77* | 0.62* | 4.30 | 11.86* | 0.83* | 0.63* | 4.25 | 2.72 | 1.04* |
| Kokkinos et al. (3)§ (mL·min⁻¹·kg⁻¹) | 0.67* | 4.94 | 14.62* | 0.88* | 0.59* | 4.66 | 15.67* | 0.81* | 0.62* | 4.30 | 11.70* | 0.88* | 0.63* | 4.25 | 2.52 | 1.09* |
| Myllus et al. (mL·min⁻¹) | 0.15* | 578 | 709.52 | 1.08* | 0.17* | 473 | 1136.18* | 0.95* | 0.14* | 465 | 1265.37* | 0.81* | 0.52* | 431 | −820.83 | 1.94* |
| Petek et al. (L·min⁻¹) | 0.12* | 0.59 | 1.23* | 0.70* | 0.14* | 0.48 | 1.57* | 0.61* | 0.15 | 0.46 | 1.31* | 0.67* | 0.50* | 0.44 | −1.80 | 1.53* |

**Males — Prior Equations in TE**

| Validated subgroup | Age 18–30 [n = 871] | | | | Age 31–45 [n = 2109] | | | | Age 46–60 [n = 333] | | | | Age ≥61‡ [n = 17] | | | |
|---|---|---|---|---|---|---|---|---|---|---|---|---|---|---|---|---|
| | R² | RMSE | C1 | C2 | R² | RMSE | C1 | C2 | R² | RMSE | C1 | C2 | R² | RMSE | C1 | C2 |
| Wilson et al.† (mL·min⁻¹·kg⁻¹) | 0.01* | 6.93 | 32.80* | 0.37* | 0.01* | 6.53 | 27.76* | 0.43* | 0.03* | 6.01 | 9.90 | 0.75* | −0.02 | 5.38 | 16.07 | 0.62 |
| Fitzgerald et al.† (mL·min⁻¹·kg⁻¹) | n/a | n/a | n/a | n/a | n/a | n/a | n/a | n/a | n/a | n/a | n/a | n/a | n/a | n/a | n/a | n/a |
| Wasserman et al. (mL·min⁻¹) | 0.27* | 492 | 1895.40* | 0.78* | 0.25* | 456 | 2124.75* | 0.71* | 0.25* | 415 | 1290.87* | 0.79* | 0.29* | 422 | 247.52 | 1.58* |
| Myers et al. (mL·min⁻¹·kg⁻¹) | 0.14* | 41.72 | 16.51* | 0.84* | 0.21* | 34.33 | 12.14* | 0.95* | 0.24* | 28.34 | 7.78 | 1.10* | −0.05 | 29.57 | 36.94 | 0.26 |
| Nevill et al. (1)§§ (mL·min⁻¹·kg⁻¹) | 0.11* | 6.56 | 34.03* | 0.45* | 0.20* | 5.87 | 22.92* | 0.70* | 0.25* | 5.30 | 16.34* | 0.91* | 0.07 | 5.13 | 21.43 | 0.74 |
| Nevill et al. (2)§§ (mL·min⁻¹·kg⁻¹) | 0.11* | 6.57 | 19.78* | 0.76* | 0.21* | 5.86 | 9.26* | 1.01* | 0.25* | 5.29 | 4.58 | 1.18* | 0.06 | 5.15 | 17.86 | 0.82 |
| Petek et al. (L·min⁻¹) | 0.08* | 0.55 | 1.03* | 0.76* | 0.17* | 0.48 | 0.07 | 0.99* | 0.20* | 0.43 | 0.10 | 0.98* | 0.50* | 0.35 | −5.01* | 2.36* |

**Females — Prior Equations in CE**

| Validated subgroup | Age 18–30‡ [n = 48] | | | | Age 31–45‡ [n = 77] | | | | Age 46–60‡ [n = 5] | | | | Age ≥61‡ [n = 0] | | | |
|---|---|---|---|---|---|---|---|---|---|---|---|---|---|---|---|---|
| | R² | RMSE | C1 | C2 | R² | RMSE | C1 | C2 | R² | RMSE | C1 | C2 | R² | RMSE | C1 | C2 |
| Wilson et al.† (mL·min⁻¹·kg⁻¹) | n/a | n/a | n/a | n/a | n/a | n/a | n/a | n/a | n/a | n/a | n/a | n/a | n/a | n/a | n/a | n/a |
| Fitzgerald et al.† (mL·min⁻¹·kg⁻¹) | −0.02 | 6.14 | 40.64 | 0.18 | 0.02 | 6.77 | 21.06 | 0.52 | 0.54 | 3.52 | 135.49* | −2.01 | n/a | n/a | n/a | n/a |
| Wasserman et al. (mL·min⁻¹) | 0.04 | 609 | 844.10 | 1.12 | 0.31* | 402 | 26.71 | 1.70* | −0.08 | 578 | −6556.75 | 6.40 | n/a | n/a | n/a | n/a |
| Kokkinos et al. (1 for males/ 2 for females)§ (mL·min⁻¹·kg⁻¹) | 0.41* | 4.68 | 23.18* | 0.57* | 0.60* | 4.36 | 11.95* | 0.82* | 0.93* | 1.35 | −34.12 | 1.94* | n/a | n/a | n/a | n/a |
| Kokkinos et al. (3)§ (mL·min⁻¹·kg⁻¹) | 0.41* | 4.68 | 23.05* | 0.61 | 0.60* | 4.36 | 11.76* | 0.87* | 0.93* | 1.35 | −34.57* | 2.07 | n/a | n/a | n/a | n/a |
| Myllus et al. (mL·min⁻¹) | 0.07* | 600 | 1093.61 | 0.91* | 0.23* | 424 | 1001.66* | 1.66* | −0.29 | 631 | 4281.22 | −1.01 | n/a | n/a | n/a | n/a |
| Petek et al. (L·min⁻¹) | 0.03 | 0.61 | 1.42 | 0.53 | 0.11* | 0.46 | 1.35* | 0.59* | −0.26* | 0.62 | 4.35 | −0.67 | n/a | n/a | n/a | n/a |

**Females — Prior Equations in TE**

| Validated subgroup | Age 18–30 [n = 233] | | | | Age 31–45 [n = 391] | | | | Age 46–60‡ [n = 46] | | | | Age ≥61‡ [n = 1] | | | |
|---|---|---|---|---|---|---|---|---|---|---|---|---|---|---|---|---|
| | R² | RMSE | C1 | C2 | R² | RMSE | C1 | C2 | R² | RMSE | C1 | C2 | R² | RMSE | C1 | C2 |
| Wilson et al.† (mL·min⁻¹·kg⁻¹) | n/a | n/a | n/a | n/a | n/a | n/a | n/a | n/a | n/a | n/a | n/a | n/a | n/a | n/a | n/a | n/a |
| Fitzgerald et al.† (mL·min⁻¹·kg⁻¹) | 0.004 | 6.61 | 48.96* | 0.02 | 0.08* | 6.19 | 8.61 | 0.77* | −0.02 | 6.73 | 34.98 | 0.22 | n/a | n/a | n/a | n/a |
| Wasserman et al. (mL·min⁻¹) | 0.27* | 393 | 40.58 | 1.56* | 0.49* | 370 | −739.77 | 2.14* | 0.03 | 372 | 1356.77 | 0.91 | n/a | n/a | n/a | n/a |
| Myers et al. (mL·min⁻¹·kg⁻¹) | 0.07* | 40.62 | 21.89* | 0.72* | 0.15* | 35.41 | 19.33* | 0.84* | 0.13* | 46.60 | 12.95 | 1.06* | n/a | n/a | n/a | n/a |
| Nevill et al. (1)§§ (mL·min⁻¹·kg⁻¹) | 0.09* | 6.29 | 27.86* | 0.56* | 0.18* | 5.84 | 22.77* | 0.74* | 0.21* | 5.91 | 9.20 | 1.21* | n/a | n/a | n/a | n/a |
| Nevill et al. (2)§§ (mL·min⁻¹·kg⁻¹) | 0.05* | 6.42 | 23.03* | 0.68* | 0.16* | 5.89 | 14.81* | 0.94* | 0.17* | 6.06 | 7.62 | 1.22* | n/a | n/a | n/a | n/a |
| Petek et al. (L·min⁻¹) | 0.18* | 0.42 | 0.45 | 0.81* | 0.32* | 0.43 | −0.37 | 1.14* | 0.10* | 0.36 | 0.61 | 0.80* | n/a | n/a | n/a | n/a |

Abbreviations: CE, cycle ergometry; R², adjusted R²; RMSE, root mean square error; C1, calibration-in-the-large; C2, calibration slope; n/a, not applicable; TE, treadmill. Comparisons between subgroups (p value) were obtained by one-way ANOVA, Student's t-test, and post-hoc HSD Tukey test. Significant values ($p < 0.05$) were marked as [*]. All values are presented in originally derived units for each model (original unit was added in bracket). †Fitzgerald et al. and Wilson et al. are meta-analyses exclusively for one sex. §Kokkinos et al. presents 3 equations for cycle ergometry: (1) only for males, (2) only for females, (3) for both males and females. §§Nevill et al. presents 2 equations for treadmill: (1) allometric model, (2) additive model. Subgroups that did not meet the TRIPOD guidelines to consider their validation results as reliable (i.e n≥100) were marked with [‡].

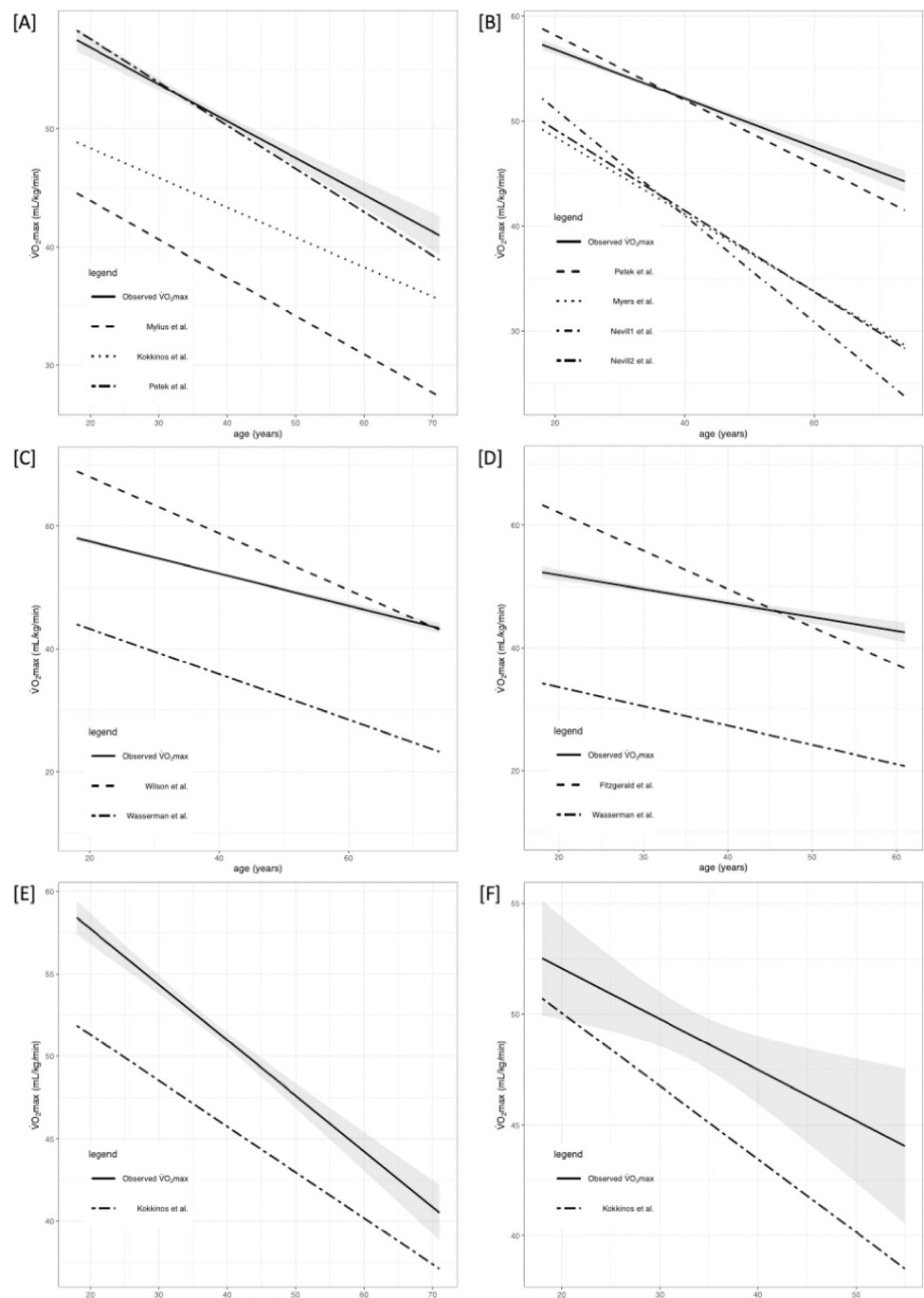

**Fig 2. Regression analysis (RA) of predicted and observed VO$_{2max}$ for whole population (both males and females with included „transition" group) stratified by age.** 95% confidence intervals are presented with grey color. Subgroups that did not meet the TRIPOD guidelines to consider their validation results as reliable (i.e n≥100) were marked with [‡]. Abbreviations: VO$_{2max}$, maximal oxygen uptake. [A] RA for males and females for CE models; [B] RA for males and females for TE models; [C] RA for males for TE or CE models (Wilson et al. and Kokkinos et al.[1]); [D] RA for females for TE or CE models (Fitzgerald et al. and Kokkinos et al.[2]); [E] RA for males for TE or CE model (Kokkinos et al.[3]); [F] RA for females for TE or CE model (Kokkinos et al.[3]).

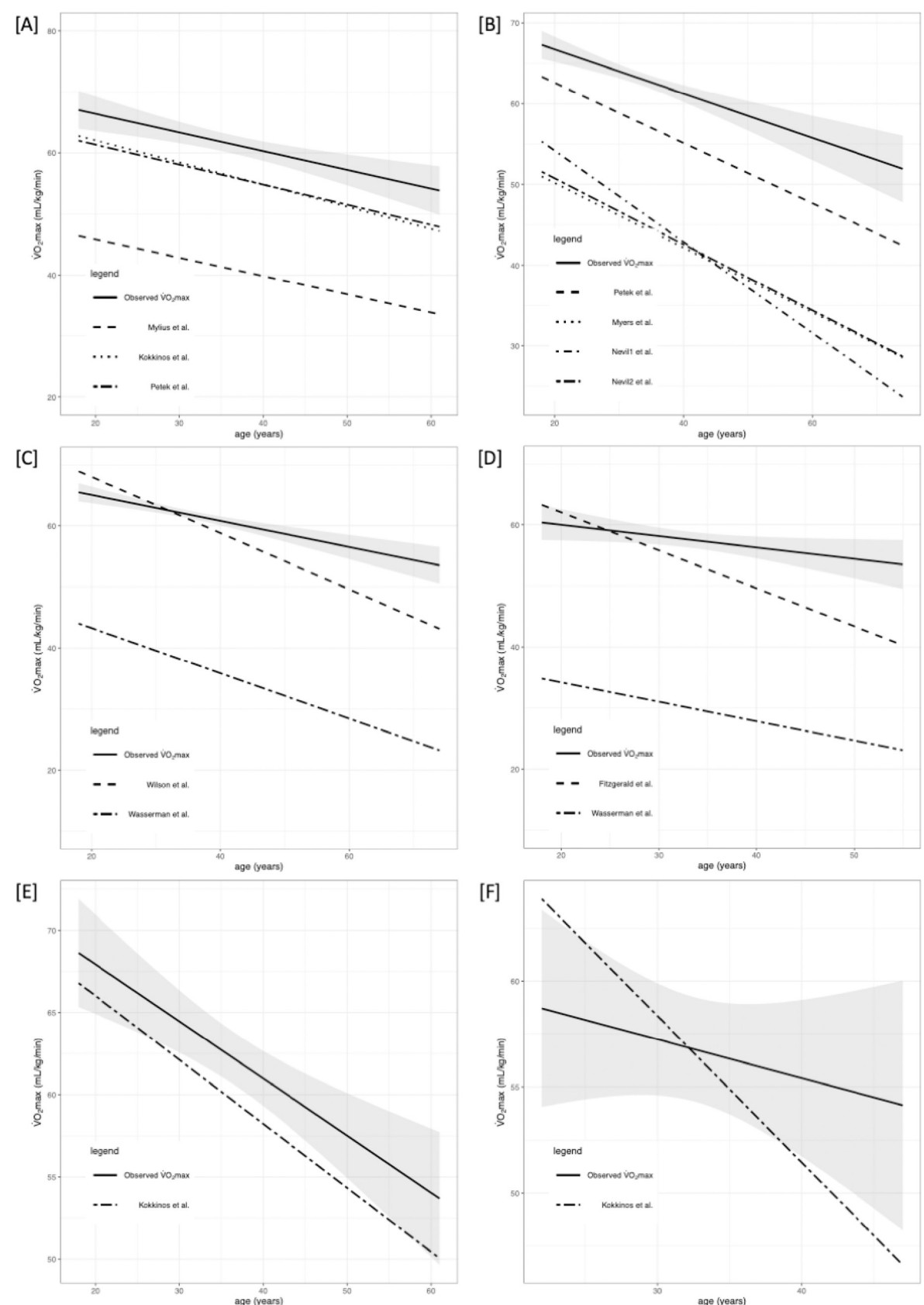

**Fig 3. Regression analysis (RA) of predicted and observed VO$_{2max}$ for HTEA stratified by age.** 95% confidence intervals are presented with grey color. Subgroups that did not meet the TRIPOD guidelines to consider their validation results as reliable (i.e n≥100) were marked with [‡]. Abbreviations: VO$_{2max}$, maximal oxygen uptake. [A] RA for males‡ and females‡ for CE models; [B] RA for males and females‡ for TE models; [C] RA for males for TE or CE‡ models (Wilson et al. and Kokkinos et al.[1]); [D] RA for females for TE or CE‡ models (Fitzgerald et al. and Kokkinos et al.[2]); [E] RA for males for TE or CE‡ model (Kokkinos et al.[3]); [F] RA for females for TE‡ or CE‡ model (Kokkinos et al.[3]).

(mean = 2.65; CI, 1.15, 4.15,). For TE, other models significantly overestimated VO$_{2max}$ (Wilson's for males and Fitzgerald's for females) or underestimated (Wasserman's, Myers's, Nevill's both allometric (1) and additive (2) formulae for males and Wasserman's, Myers's, Nevill's (1) and (2) for females). For CE, significant overestimation was observed in Wilson's and Fitzgerald's models respectively for males and females, and underestimation in Wasserman's, Mylius's, and Kokkinos's formulae for both males, females, and the whole population.

For HTEA a group size ≥ 100 was only for male runners (n = 188). For TE, significant differences between the observed and predicted VO$_{2max}$ were both in males and females for the HTEA subgroup. The lowest obtained differences on TE were for the Wilson's for males (mean = 2.52; CI, 1.50, 3.54) and for the Fitzgerald's for females (mean = 4.1; CI, 1.92, 6.28). For CE, there were no significant differences for Wilson's model (mean = 1.82; CI, –0.32, 3.96) and Fitzgerald's model formula (mean = 2.57; CI, –1.38, 6.52), respectively for male and female EA.

The equation based on the general population explained varied performance, from none or very weak effect size, up to moderate. For TE, $R^2$ ranged from 0.28 for Nevill's (1) equation up to 0.54 for Petek's equation. In CE, $R^2$ ranged from 0.38 for Petek's equation up to 0.64 in Kokkinos's equations. In the HTEA cohort (only one with n≥100), for males TE, $R^2$ ranged from 0.15 in Wilson's equation up to 0.44 in Wasserman's equation. Although, they were poorly calibrated (for Wasserman's C1 = 1461.98 mL·min$^{-1}$).

## Discussion

The aim of the current study was to assess the accuracy of common VO$_{2max}$ prediction equations in a large sample of healthy EA tested under standardized conditions. We hypothesized that their accuracy may not be adequate to make them a comparable approach for CPET.

The main novelty of this study is a comprehensive comparison of the accuracy of various formulas and their usefulness for determining VO$_{2max}$ in the athletic population (including different subpopulations depending on participants' training level). The analysis of the accuracy of prediction equations suggests that more precise models are required to better establish the VO$_{2max}$ level, which may be crucial for the clinical assessment of EAs.

The main findings are that: (1) the currently available equations show limited accuracy, (2) it is most recommended to use models derived from populations with the possibly most similar characteristics to the target group, (3) models derived from active athletic populations works the most accurately and showed the highest transferability, and (4) a steeper decline in predicted VO$_{2max}$ for older participants was noted.

### Current limitations in model's transferability

Until now, most frequently underestimation of results of what in younger EA and overestimation in older ones have been observed [21,29]. Malek et al. found that 16 of 18 commonly used prediction equations were inaccurate when used in an athlete population [29]. Moreover, there was a lack of equations to predict VO$_{2max}$ developed in large samples of trained participants, especially elite athletes [21]. In one recent study, Petek et. al. validated previous and

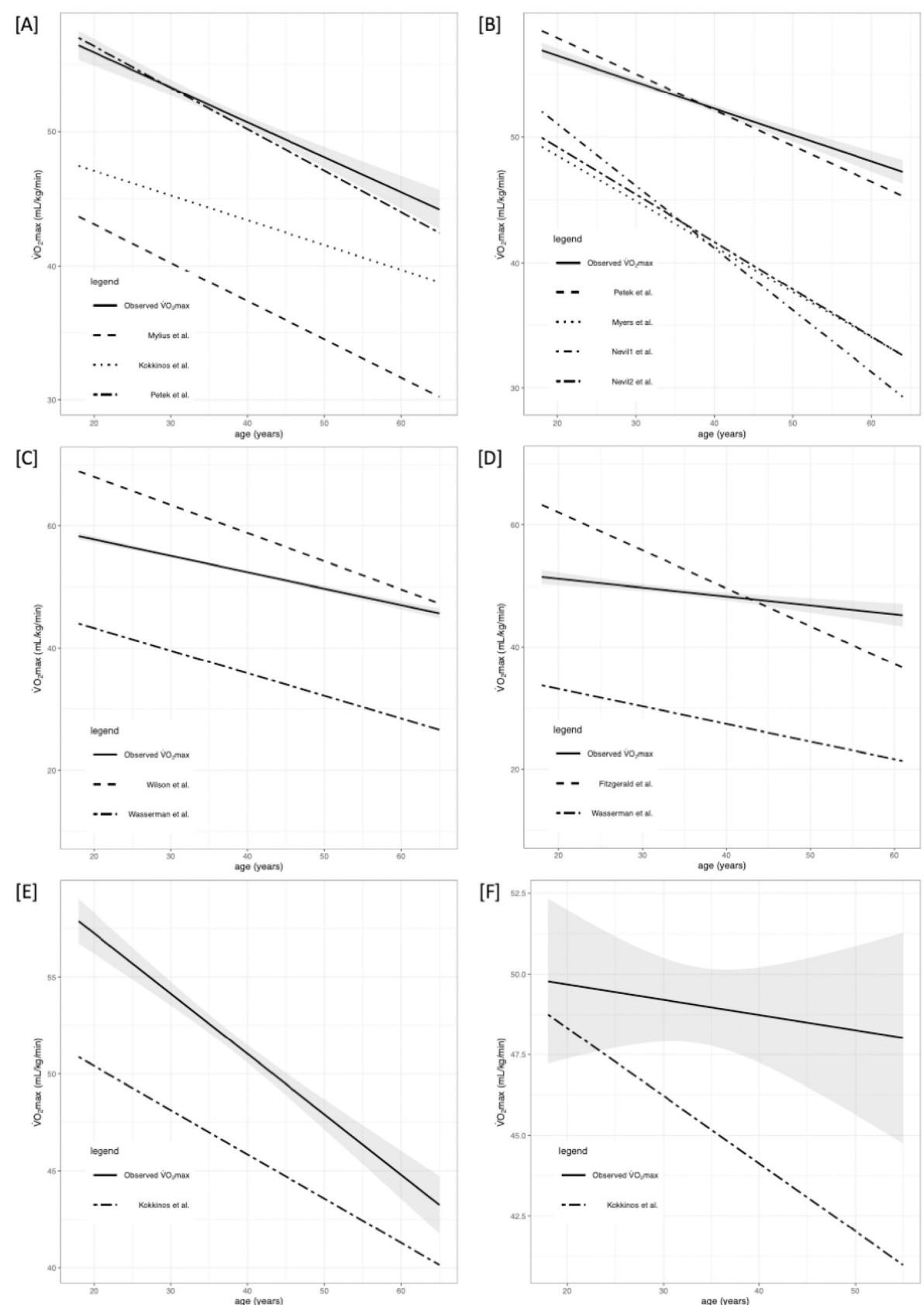

**Fig 4. Regression analysis (RA) of predicted and observed VO$_{2max}$ for REA stratified by age.** 95% confidence intervals are presented with grey color. Subgroups that did not meet the TRIPOD guidelines to consider their validation results as reliable (i.e n≥100) were marked with [‡]. Abbreviations: VO$_{2max}$, maximal oxygen uptake. [A] RA for males and females‡ for CE models; [B] RA for males and females‡ for TE models; [C] RA for males for TE or CE models (Wilson et al. and Kokkinos et al.[1]); [D] RA for females for TE‡ or CE‡ models (Fitzgerald et al. and Kokkinos et al.[1]); [E] RA for males for TE or CE model (Kokkinos et al.[3]); [F] RA for females for TE‡ or CE‡ model (Kokkinos et al.[3]).

developed new VO$_{2max}$ equations for EA, although the sample size was relatively small [21]. Their main finding was that the previously established models, both on general cohorts and EA, perform poorly when used for EA undergoing CPET for clinical reasons.

Valid VO$_{2max}$ prediction equations are important as they can lead to false-negative or false-positive results and inadequate recommendations regarding a safe level of physical activity or the level of advancement of the training plan [8,21]. Furthermore, the normality of the VO$_{2max}$ values is often a very important step to determine the cause of the exercise limitation [53].

Practical application of the most accurately derived predictive equations is a better distinction of physiological vs impaired endurance. Moreover, it undoubtedly improves the clinical usage of VO$_{2max}$ assessment for EA examined with the suspected or confirmed CVD or to precisely prepare individualized training plans.

One of the reasons for obtaining very heterogeneous predicted results is the discrepancy in the methodology [21,30]. Potential complications were mainly related to CPET- usage of cardiology-specific protocols for TE (e.g., Bruce protocol [54]) which are not commonly used in sports-performance diagnostics [55]. Individual running economy, general fatigue, or nonspecific stress during testing rise the probability of bias [20]. The error could be even up to 40% of the actual value [28]. Our study population is larger and contains EA from the individual- or team-sports disciplines. The testing protocol was strictly standardized, and measurements included advanced parameters influencing performance- LA and BM [55].

## Repercussions of using predictive equations with low to moderate accuracy

Consequences of applying prediction models with limited accuracy could be seen in sports and performance diagnostics, clinical practice, the applicability of particular equations, training prescription, and follow-up. In sports diagnostics, this can lead to prescribing incorrect, ineffective training [56]. In sports medicine and cardiology obtaining accurate VO$_{2max}$ values is especially important for patients with CVDs, given the growing data suggesting the role of CRF in stratifying the risk in such groups [3,57]. Moreover, there are currently more inaccuracies in the estimation of VO$_{2max}$ among patients suffering from CVDs than in healthy individuals. Overestimation is especially noticeable among patients with impaired cardiac output during exercise. Relying on inaccurate VO$_{2max}$ values may result in a missed diagnosis and incorrectly prescribed therapy, which does not bring the expected results and poses a health risk. Among other conditions, of particular importance is heart failure as referred to by Kokkinos et al [57]. Furtherly, for such equations, their applicability is limited to narrow populations with characteristics as close as possible to the group from which they were originally created (i.e. derivation cohorts) [21].

## Specificity of particular subgroups

Outcome of the present study was that the examined prediction equations of VO2max had limited prediction value in the locomotion (running versus cycling), age, and performance subgroups of participants. An explanation of this limited prediction value might be due to the selection of specific predictors (sex, age, and weight) that were not measures of CRF. Among

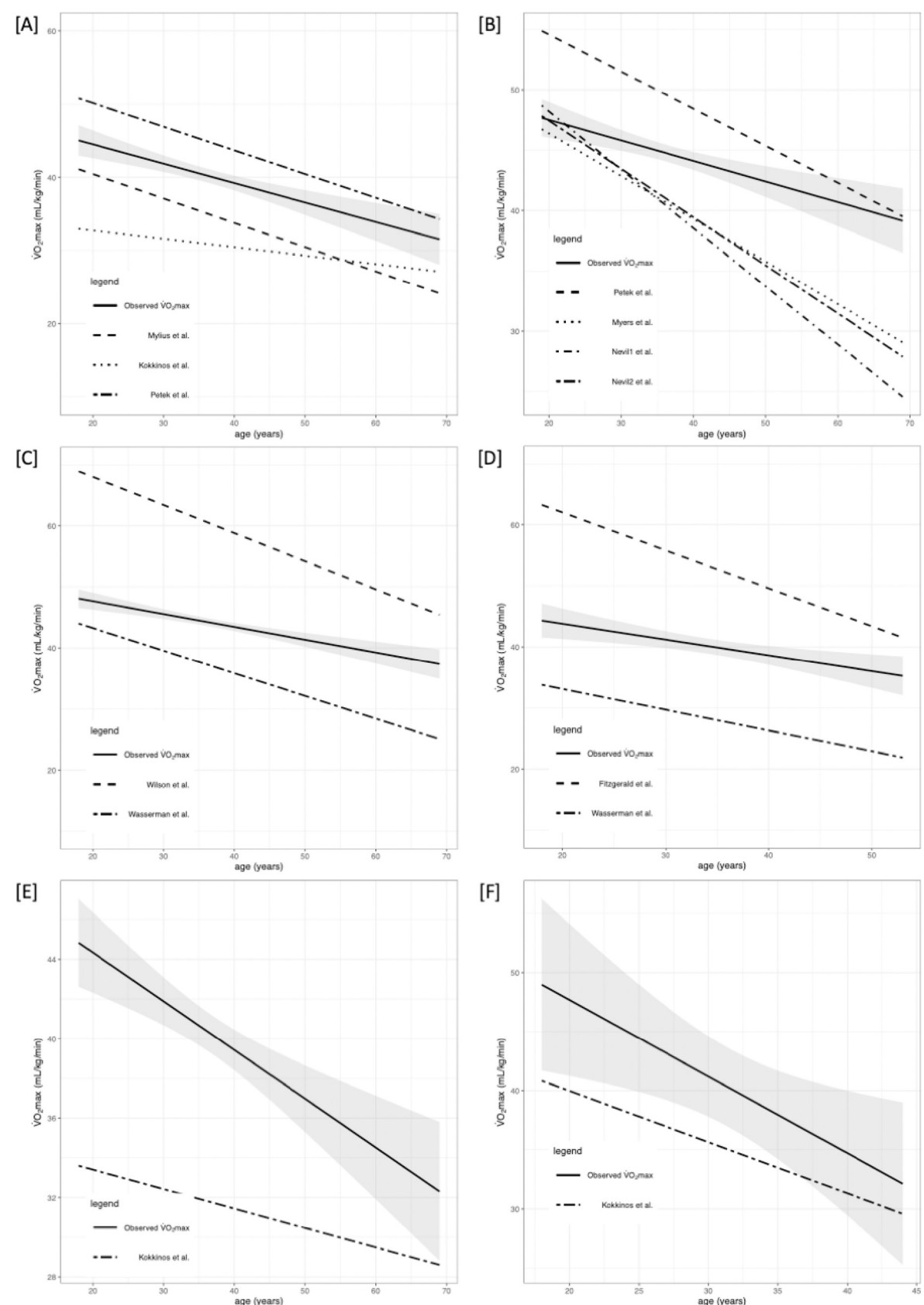

**Fig 5. Regression analysis (RA) of predicted and observed VO$_{2max}$ for LTEA stratified by age.** 95% confidence intervals are presented with grey color. Subgroups that did not meet the TRIPOD guidelines to consider their validation results as reliable (i.e n≥100) were marked with [‡]. Abbreviations: VO$_{2max}$, maximal oxygen uptake. [A] RA for males‡ and females‡ for CE models; [B] RA for males and females for TE models; [C] RA for males for TE or CE‡ models (Wilson et al. and Kokkinos et al.[1]); [D] RA for females for TE‡ or CE‡ models (Fitzgerald et al. and Kokkinos et al.[2]); [E] RA for males for TE or CE‡ model (Kokkinos et al.[3]); [F] RA for females for TE‡ or CE‡ model (Kokkinos et al.[3]).

the selected predictors, the only mechanical workload was a measure of CRF. CRF in EA consisted not only of a health-related but also a sport-related physical fitness parameter; thus, it would be of great practical importance that predicted VO2max could reflect changes in sports performance. Furthermore, performance subgroups of participants might differ for body composition (i.e. lower body fat percentage in HTEA than in LTEA), which in turn might consist a bias in the assessment of CRF [58]. Ceaser and Hunter point out that endurance capacity may also depend on participant ethnicity, so this factor should be considered when deriving new models [18,58]. It is worth noting that models derived from wide populations or EA groups showed the highest performance. This is in line with the results observed so far by Petek et al. and Malek et al. [21,29]. The sex-specific equations (i.e. those provided by Kokkinos et al for CE) did not show noticeable higher accuracy. The underlying mechanism remains further investigation as females are presented physiologically with lower VO$_{2max}$ than males [59]. A steeper decline in predicted values for older EA in VO$_{2max}$ may be justified by maintaining a higher VO$_2$ with age through regular physical activity. Similar results have already been confirmed by Kaminsky et al [59]. EAs observe a lower decline in VO$_2$ with age than their corresponding reference group [59]. The highest inaccuracies have been noted for HTEA and younger participants. Perhaps, due to their physically higher VO$_{2max}$ which is also supported by additional endurance training. Thus, those EA placed above normal reference values [59]. As we underlined, there is a need for more advanced prediction models which will consider additional parameters (like age and physical activity) and fits demanding of HTEA and young individuals.

## Source of errors measured & differences between protocols

The observed R$^2$ ranged from 0.02 (Fitzgerald et al. equation for TE) to 0.65 (Kokkinos et al. equation for CE). Large discrepancies between particular equations were observed. Briefly, RMSE support the equation provided by Petek et al. The one for TE represented the lowest RMSE from all validated formulae (RMSE = 0.48). We stipulate that this finding could result from similar characteristics of the primary cohort to our validation cohort. As different CPET results are achieved on CE and TE, the Petek et al. models adjusted for both modalities and athletic group showed the best performance. It is worth mentioning, that the tested formulae represented generally lower validity for women. It is well established that female athletes achieve lower CPET scores compared to male athletes. Although, the underlying mechanism for the reduced VO$_2$ prediction rate in this sex remains unclear. Moreover, the results between the models differed significantly (p<0.05), despite validating them on our one population. It is suggested to consider the most outlying results. More research is needed to refine the effects and recalibration of the currently available equations. The models derived from broader cohorts (provided from FRIENDS by Kokkinos et al.) or sports cohorts (provided by Petek et al.) showed less inaccuracy in both direct comparison of measured and predicted VO$_2$ and statistical indices (R$^2$, RMSE and calibrations). Thus, we recommended them for estimating V̇O$_{2max}$ in male and female EAs. We would like to note that the equations derived from meta-analyses, i.e. Figerald and Wilson represented the smallest inaccuracy between directly measured VO$_{2max}$ and predicted. The salient feature of the meta-analysis equations is that they

utilize attainable demographic information for the widest cohorts. This advantage is not feasible for equations derived from original papers due to practical limitations in recruiting such a numerous amount of participants. To summarize, models represent wide differences, and innacuracies were lower when applied to cohorts with comparable profiles.

### Directions of future research

We recommend that the formulas used to estimate $VO_{2max}$ should be applied to groups with a similar profile to the one from which they were originally derived, especially in narrow populations like LTEA, REA, or HTEA [45]. At the same time, we emphasize that there is a significant need to create new, more advanced models under unified guidelines and with the incorporation of PROBAST-AI [60] and TRIPOD checklist [35]. It will facilitate the further selection of the appropriate equation to apply in EA depending on their level of CRF. In addition, the need of selecting other predictors, such as oxygen uptake at submaximal exercise intensity, ethnicity, or a daily number of steps, should be considered in future studies.

### Conclusions

To conclude, we have accomplished an independent external validation of prognostic models for the prediction of the CRF level, defined as a $VO_{2max}$. Each included prognostic model showed only moderate discriminatory ability, but acceptable performance at derivation population. Direct $VO_{2max}$ determination by CPET cannot be replaced or interchangeable with predictive equations for EA based only on their own results. An updated and unified prognostic formula for clinical and experimental use in EA populations is necessary. Despite no formula being completely exact, the best performance was noted for males on the CE in Kokkinos model ($R^2 = 0.64$) and males on the TE in the Wasserman model ($R^2 = 0.26$), whereas for females on the CE in Kokkinos ($R^2 = 0.65$) and female on the TE in Wasserman ($R^2 = 0.30$) equations. Those models seem to better predict $VO_{2max}$ in our EA population and may provide utility as a method-of-choice in assessment tool during sports diagnostics or clinical practice. The overall lowest model accuracy has been observed for HTEA and EA 18–30 yr. A potential limitation of the study was the ethnic homogeneity of our group, as the subjects were mainly Caucasian.

### Supporting information

**S1 File. Cardiopulmonary exercise.**
(DOCX)

**S2 File. Prediction equations.**
(DOCX)

**S3 File. CPET characteristics.**
(DOCX)

**S4 File. Additional models.**
(DOCX)

**S5 File. Tripod checklists.**
(DOCX)

### Author Contributions

**Conceptualization:** Szczepan Wiecha.

**Data curation:** Szczepan Wiecha, Igor Cieśliński.

**Formal analysis:** Szczepan Wiecha, Tomasz Palka, Beat Knechtle, Daniel Śliż.

**Investigation:** Szczepan Wiecha.

**Methodology:** Szczepan Wiecha, Beat Knechtle, Pantelis T. Nikolaidis, Łukasz A. Małek.

**Project administration:** Szczepan Wiecha, Przemysław Seweryn Kasiak, Daniel Śliż.

**Resources:** Szczepan Wiecha.

**Software:** Igor Cieśliński.

**Supervision:** Tim Takken, Tomasz Palka, Marek Postuła, Artur Mamcarz.

**Validation:** Szczepan Wiecha.

**Visualization:** Igor Cieśliński.

**Writing – original draft:** Szczepan Wiecha, Przemysław Seweryn Kasiak, Pantelis T. Nikolaidis, Daniel Śliż.

**Writing – review & editing:** Szczepan Wiecha, Tim Takken, Beat Knechtle, Łukasz A. Małek.

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
