## [Decision Letter · Decision Letter 0]

10 Aug 2022

PONE-D-22-16633External validation of VO2max prediction models based on recreational and elite endurance athletesPLOS ONE

Dear Dr. Wiecha,

Thank you for submitting your manuscript to PLOS ONE. After careful consideration, we feel that it has merit but does not fully meet PLOS ONE’s publication criteria as it currently stands. Therefore, we invite you to submit a revised version of the manuscript that addresses the points raised during the review process.

Both reviewers agree that there is a weakness in the reasons and rationale that supports the work. What are the novelties of the manuscript?

I recommend a comprehensive review covering every aspect highlighted by the reviewers.

We look forward to receiving your revised manuscript.

Kind regards,

Alvaro Reischak-Oliveira, Ph.D.

Academic Editor

PLOS ONE

Journal Requirements:

3. Please upload a copy of Figure 1, to which you refer in your text on page 15. If the figure is no longer to be included as part of the submission please remove all reference to it within the text.

Reviewers' comments:

Reviewer's Responses to Questions

**Comments to the Author**

1. Is the manuscript technically sound, and do the data support the conclusions?

Reviewer #1: Yes

Reviewer #2: Partly

2. Has the statistical analysis been performed appropriately and rigorously? 

Reviewer #1: Yes

Reviewer #2: I Don't Know

3. Have the authors made all data underlying the findings in their manuscript fully available?

Reviewer #1: Yes

Reviewer #2: Yes

4. Is the manuscript presented in an intelligible fashion and written in standard English?

Reviewer #1: Yes

Reviewer #2: Yes

5. Review Comments to the Author

Reviewer #1: General Comments

The manuscript provides information on external validation of VO2max prediction models based on recreational and elite endurance athletes. Overall, the study is well-conducted, methodologically robust, and provides relevant information for the area of sports and health sciences. My concern about the study is mainly related to the introduction, methodology, and discussion sections, which could be edited. Please, see in the specific comments.

Specific Comments

Personal suggestion: I believe the main message of the study is that direct VO2max determination (CPET plus ergospirometry) cannot be replaced or interchangeable by predictive equations for endurance athletes based on their own results 0.30 < R2 < 0.65. If the authors agree with me, please make this message clearer in the discussion and conclusion of the study. Additionally, little is discussed about the repercussions of using predictive equations with low to moderate accuracy for the contexts of sports & performance diagnostics, clinical practice, applicability of prediction equation, training prescription and follow-up.

Abstract:

What is the criterion used to define models with the highest accuracy?

Please, include the minimum and maximum values for underestimated and overestimated that can be obtained for indirect VO2max.

Introduction: The introduction does not convince the reader of the importance of direct determination of VO2max compared to indirect determination through equations. Additionally, the authors could include epidemiological data to strengthen the argument for the importance of VO2max for the health of endurance athletes. Please make it clear in the introduction what is the main focus and secondary outcomes of the study. In addition, to include the advantages and disadvantages of each of the ways of determining VO2max for the contexts of sports & performance diagnostics, clinical practice, applicability of prediction equation, training prescription and follow-up.

Lines 18-19: Please, describe more variables that can influence the variability of VO2max data in endurance athletes.

Lines 33-34: Please, deepen this information, including epidemiological data, if possible, on CVDs in endurance athletes. Please, include information here about the importance of directly measured VO2max can be used as a predictive cardiometabolic risk factor. In addition, there is little in the introduction about the limitations or harm that the indirect determination of VO2max may have in relation to clinical evaluation. Please, present the risks and benefits of direct and indirect determination (prediction equations) of the VO2max data.

Line 55: There is little information about why endurance athletes were chosen as a sample for the study.

Methodology:

Lines 79-80: Please make it clear whether data collection was performed with a breath-by-breath acquisition system and a 10-seconds filter was used for data analysis. Note that this information can lead to some confusion as to how VO2max was defined as described in line 104 “Maximal oxygen uptake (VO2max) was defined as an averaged maximum oxygen uptake during the 15-s period at the end of the CPET’. Could I think that only 1 or 2 VO2 points were analysed for the average of VO2max? Please clarify.

Suggestion: Please, insert the 13 equations of selected predictions models to establish pVO2max in methodology section.

Statistical Analysis: Please, describe the criterion used to classify low to moderate variability in VO2max. What is the criterion used to classify accuracy (low-moderate-high)? Please, describe.

Discussion: The authors could better highlight the main findings of the study. In addition, authors should focus the discussion section on possible explanation of their main results of the present study.

Suggestion: The authors could organize the discussion as follows: Describe the consequences (risks and benefits) of using VO2max prediction models with low to moderate accuracy (underestimation and overestimation values) in the following contexts: (1) Sports & Performance Diagnostics; (2) Clinical Practice; (3) Endurance Training Prescription and Follow-Up.

Reviewer #2: I don't understand very well this study. If you check the correlation between a previously validated formula in very specific conditions with actual values in another different setting, it is obvious that the results will be quite different. The authors should guarantee that the comparisons are made with the same protocols (step duration, starting intensity, exhaustion criteria, etc.) and populations (age, sex, sport, training level, etc.) between the current and previous validation studies.

The figures are not legible.

VO2max can be tested on the field in a sport setting. An example:

https://pubmed.ncbi.nlm.nih.gov/24936896/

6. PLOS authors have the option to publish the peer review history of their article (what does this mean?). If published, this will include your full peer review and any attached files.

Reviewer #1: **Yes: **Dr. Giovani dos Santos Cunha

Reviewer #2: No

---

## [Author Response · Author response to Decision Letter 0]

30 Sep 2022

Author’s response to reviews

Dear reviewers,

We are grateful for the time and effort you took to write these extensive reviews. We have done our best to incorporate all of your suggestions and send back the revised manuscript. We believe the content is much improved and more clear for readers thanks to your comments.

Yours sinserely,

Szczepan Wiecha on behalf of all authors.

In the following, please find our answers to your comments. 

Title: External validation of VO2max prediction models based on recreational and elite

endurance athletes

Version: 1 Date: 30 Aug 2022

In the following, please find our answers to your comments. 

Reviewers comments: 

Dr. Giovani dos Santos Cunha (Reviewer 1)

General Comments

The manuscript provides information on external validation of VO2max prediction models based on recreational and elite endurance athletes. Overall, the study is well-conducted, methodologically robust, and provides relevant information for the area of sports and health sciences. My concern about the study is mainly related to the introduction, methodology, and discussion sections, which could be edited. Please, see in the specific comments.

Specific Comments

Personal suggestion: I believe the main message of the study is that direct VO2max determination (CPET plus ergospirometry) cannot be replaced or interchangeable by predictive equations for endurance athletes based on their own results 0.30 < R2 < 0.65. If the authors agree with me, please make this message clearer in the discussion and conclusion of the study. Additionally, little is discussed about the repercussions of using predictive equations with low to moderate accuracy for the contexts of sports & performance diagnostics, clinical practice, applicability of prediction equation, training prescription and follow-up .

Response: Thank you for the suggestion, we have added this information to the introduction section.

Abstract:

What is the criterion used to define models with the highest accuracy? 

Response: We added information: “by assessing the transferability of the currently available prediction models based on their R2, calibration-in-the-large, and calibration slope”

Please, include the minimum and maximum values for underestimated and overestimated that can be obtained for indirect VO2max. 

Response: Due to the limited capacity of the abstract, we are not able to provide as much detail for each of the validated equations. These details are given in the results section and the supplement

Introduction: The introduction does not convince the reader of the importance of direct determination of VO2max compared to indirect determination through equations. Additionally, the authors could include epidemiological data to strengthen the argument for the importance of VO2max for the health of endurance athletes. Please make it clear in the introduction what is the main focus and secondary outcomes of the study. In addition, to include the advantages and disadvantages of each of the ways of determining VO2max for the contexts of sports & performance diagnostics, clinical practice, applicability of prediction equation, training prescription and follow-up .

Response: Thank you very much for your insightful analysis of the content. We have amended the content in line with your suggestions

Lines 18-19: Please, describe more variables that can influence the variability of VO2max data in endurance athletes.

Response: We added additional variables as suggested

Lines 33-34: Please, deepen this information, including epidemiological data, if possible, on CVDs in endurance athletes. Please, include information here about the importance of directly measured VO2max can be used as a predictive cardiometabolic risk factor. In addition, there is little in the introduction about the limitations or harm that the indirect determination of VO2max may have in relation to clinical evaluation. Please, present the risks and benefits of direct and indirect determination (prediction equations) of the VO2max data. 

Response: We have added more detailed information including epidemiology and clinical evaluation

Line 55: There is little information about why endurance athletes were chosen as a sample for the study .

Response: We have added information in this regard

Methodology:

Lines 79-80: Please make it clear whether data collection was performed with a breath-by-breath acquisition system and a 10-seconds filter was used for data analysis. Note that this information can lead to some confusion as to how VO2max was defined as described in line 104 “Maximal oxygen uptake (VO2max) was defined as an averaged maximum oxygen uptake during the 15-s period at the end of the CPET’. Could I think that only 1 or 2 VO2 points were analyzed for the average of VO2max? Please clarify .K 

Response: We have corrected this section. Measurements were taken for each breath and averaging was done at 15-second intervals. The 10-second values were entered by mistake. Many thanks for finding this inaccuracy in the text

Suggestion: Please, insert the 13 equations of selected predictions models to establish pVO2max in methodology section.

Response: Thank you for your suggestion but we do not want to duplicate content. A detailed table describing all included equations can be found in the manuscript supplement. In the main body of the paper, such a table would take up a considerable amount of space. Bibliographic data for the included designs are also included in the supplement

Statistical Analysis: Please, describe the criterion used to classify low to moderate variability in VO2max. What is the criterion used to classify accuracy (low-moderate-high)? Please, describe 

Response: The criterion for determining the accuracy of the models was R2 and RMSE, as described in lines 173-183. We did not use cut-off points for these parameters, as there is no explicit definition adopted here as in, for example, correlation coefficients or effect measures.

Discussion: The authors could better highlight the main findings of the study. In addition, authors should focus the discussion section on possible explanation of their main results of the present study. 

Suggestion: The authors could organize the discussion as follows: Describe the consequences (risks and benefits) of using VO2max prediction models with low to moderate accuracy (underestimation and overestimation values) in the following contexts: (1) Sports & Performance Diagnostics; (2) Clinical Practice; (3) Endurance Training Prescription and Follow-Up. 

Response: We have made amendments

Reviewer #2: I don't understand very well this study. If you check the correlation between a previously validated formula in very specific conditions with actual values in another different setting, it is obvious that the results will be quite different. The authors should guarantee that the comparisons are made with the same protocols (step duration, starting intensity, exhaustion criteria, etc.) and populations (age, sex, sport, training level, etc.) between the current and previous validation studies

Response: We have added appropriate explanations in the text, mainly in the introduction of the manuscript and methodology regarding the need for external validation and differences in protocols and VO2max values obtained. By following the guidelines for validation studies (such as the TRIPOD statement), we obtained a cross-sectional analysis of the available equations, which enabled us to reach appropriate conclusions.

The figures are not legible. 

Response: This is most likely a matter of converting the document in the editorial system for review purposes. The uploaded charts are in the correct high resolution and size. In the PLOS submissions system, they are presented on A4 size, each on a separate page. In the top right corner of the manuscript there is a hyperlink to download the TIFF version. A higher resolution version is then downloaded

VO2max can be tested on the field in a sport setting. An example:

https://pubmed.ncbi.nlm.nih.gov/24936896/

Response: Thank you very much for your valuable publication, we have completed the information

Once more, we would like to thank the Reviewers for their important comments.

---

## [Decision Letter · Decision Letter 1]

12 Dec 2022

PONE-D-22-16633R1External validation of VO2max prediction models based on recreational and elite endurance athletesPLOS ONE

Dear Dr. Wiecha,

Thank you for submitting your manuscript to PLOS ONE. After careful consideration, we feel that it has merit but does not fully meet PLOS ONE’s publication criteria as it currently stands. Therefore, we invite you to submit a revised version of the manuscript that addresses the points raised during the review process.

First of all, apologies for the time elapsed. it was somewhat difficult to get all the necessary revisions to a decision. Please pay attention to reviewer 2. Please, elaborate better on the error measures reported, the potential sources of these errors, and the differences between protocols and formulae in the discussion section.

I believe that this detail is sufficient for a final decision. Please submit your revised manuscript by Jan 26 2023 11:59PM. If you will need more time than this to complete your revisions, please reply to this message or contact the journal office at plosone@plos.org. Please include the following items when submitting your revised manuscript:A rebuttal letter that responds to each point raised by the academic editor and reviewer(s). You should upload this letter as a separate file labeled 'Response to Reviewers'.A marked-up copy of your manuscript that highlights changes made to the original version. You should upload this as a separate file labeled 'Revised Manuscript with Track Changes'.An unmarked version of your revised paper without tracked changes. You should upload this as a separate file labeled 'Manuscript'.If applicable, we recommend that you deposit your laboratory protocols in protocols.io to enhance the reproducibility of your results. Protocols.io assigns your protocol its own identifier (DOI) so that it can be cited independently in the future. For instructions see: https://journals.plos.org/plosone/s/submission-guidelines#loc-laboratory-protocols. Additionally, PLOS ONE offers an option for publishing peer-reviewed Lab Protocol articles, which describe protocols hosted on protocols.io. Read more information on sharing protocols at https://plos.org/protocols?utm_medium=editorial-email&utm_source=authorletters&utm_campaign=protocols.

We look forward to receiving your revised manuscript.

Kind regards,

Alvaro Reischak-Oliveira, Ph.D.

Academic Editor

PLOS ONE

Journal Requirements:

Reviewers' comments:

Reviewer's Responses to Questions

**Comments to the Author**

1. If the authors have adequately addressed your comments raised in a previous round of review and you feel that this manuscript is now acceptable for publication, you may indicate that here to bypass the “Comments to the Author” section, enter your conflict of interest statement in the “Confidential to Editor” section, and submit your "Accept" recommendation.

Reviewer #1: All comments have been addressed

Reviewer #2: All comments have been addressed

2. Is the manuscript technically sound, and do the data support the conclusions?

Reviewer #1: Yes

Reviewer #2: Yes

3. Has the statistical analysis been performed appropriately and rigorously? 

Reviewer #1: Yes

Reviewer #2: Yes

4. Have the authors made all data underlying the findings in their manuscript fully available?

Reviewer #1: Yes

Reviewer #2: Yes

5. Is the manuscript presented in an intelligible fashion and written in standard English?

Reviewer #1: Yes

Reviewer #2: Yes

6. Review Comments to the Author

Reviewer #1: (No Response)

Reviewer #2: Please, elaborate better on the error measures reported, the potential sources of these errors, and the differences among protocols and formulae in the discussion section.

7. PLOS authors have the option to publish the peer review history of their article (what does this mean?). If published, this will include your full peer review and any attached files.

Reviewer #1: **Yes: **Dr. Giovani dos Santos Cunha

Reviewer #2: **Yes: **Daniel Boullosa

---

## [Author Response · Author response to Decision Letter 1]

16 Dec 2022

Dear Editor, 

Dear Reviewers, 

Thank you very much for the thorough analysis of our manuscript, for your valuable and helpful comments and for giving us the opportunity to revise and improve our submission. We hope that our replies and explanations, as well as the amendments to the manuscript, fully address your concerns. We keep the change tracking. In the following, please find our answers to your comments.

Reviewer #2: Please, elaborate better on the error measures reported, the potential sources of these errors, and the differences among protocols and formulae in the discussion section.

Response: We have added a new section to the discussion

Once more, we would like to thank the Reviewer for important comments and for input on our report. We believe that all remaining concerns are now fully addressed.

With regards

Szczepan Wiecha, PhD

---

## [Decision Letter · Decision Letter 2]

11 Jan 2023

External validation of VO2max prediction models based on recreational and elite endurance athletes

PONE-D-22-16633R2

Dear Dr. Wiecha,

We’re pleased to inform you that your manuscript has been judged scientifically suitable for publication and will be formally accepted for publication once it meets all outstanding technical requirements.

Kind regards,

Alvaro Reischak-Oliveira, Ph.D.

Academic Editor

PLOS ONE

Reviewers' comments:

Reviewer's Responses to Questions

**Comments to the Author**

1. If the authors have adequately addressed your comments raised in a previous round of review and you feel that this manuscript is now acceptable for publication, you may indicate that here to bypass the “Comments to the Author” section, enter your conflict of interest statement in the “Confidential to Editor” section, and submit your "Accept" recommendation.

Reviewer #1: All comments have been addressed

Reviewer #2: All comments have been addressed

2. Is the manuscript technically sound, and do the data support the conclusions?

Reviewer #1: (No Response)

Reviewer #2: Yes

3. Has the statistical analysis been performed appropriately and rigorously? 

Reviewer #1: (No Response)

Reviewer #2: Yes

4. Have the authors made all data underlying the findings in their manuscript fully available?

Reviewer #1: (No Response)

Reviewer #2: Yes

5. Is the manuscript presented in an intelligible fashion and written in standard English?

Reviewer #1: (No Response)

Reviewer #2: Yes

6. Review Comments to the Author

Reviewer #1: (No Response)

Reviewer #2: No more comments. Although the writing may be improved, I think this manuscript adds relevant information in the context of the current evidence.

No more comments. Although the writing may be improved, I think this manuscript adds relevant information in the context of the current evidence.

7. PLOS authors have the option to publish the peer review history of their article (what does this mean?). If published, this will include your full peer review and any attached files.

Reviewer #1: **Yes: **Dr. Giovani dos Santos Cunha

Reviewer #2: **Yes: **Daniel Boullosa

---

## [Editor Report · Acceptance letter]

16 Jan 2023

PONE-D-22-16633R2 

External validation of VO_2max_ prediction models based on recreational and elite endurance athletes 

Dear Dr. Wiecha:

I'm pleased to inform you that your manuscript has been deemed suitable for publication in PLOS ONE. Congratulations! Your manuscript is now with our production department. 

Kind regards, 

on behalf of

Dr. Alvaro Reischak-Oliveira 

Academic Editor

PLOS ONE